# The Impact of Partial Splenic Embolization on Portal Hypertensive Gastropathy in Cirrhotic Patients with Portal Hypertension

**DOI:** 10.3390/jcm12072662

**Published:** 2023-04-03

**Authors:** Michio Saeki, Hironao Okubo, Yusuke Takasaki, Eisuke Nakadera, Yuka Fukuo, Hiroo Fukada, Yuta Hotchi, Hitoshi Maruyama, Shigehiro Kokubu, Shuichiro Shiina, Akihito Nagahara, Kenichi Ikejima

**Affiliations:** 1Department of Gastroenterology, Juntendo University Nerima Hospital, Tokyo 177-8521, Japan; m-saeki@juntendo.ac.jp (M.S.); ytakasa@juntendo.ac.jp (Y.T.); enakader@juntendo.ac.jp (E.N.); yfukuo@juntendo.ac.jp (Y.F.); shkokubu@gmail.com (S.K.); 2Department of Gastroenterology, Graduate School of Medicine, Juntendo University, Tokyo 113-8421, Japan; h-fukada@juntendo.ac.jp (H.F.); h.maruyama.tw@juntendo.ac.jp (H.M.); shiinas@juntendo.ac.jp (S.S.); nagahara@juntendo.ac.jp (A.N.); ikejima@juntendo.ac.jp (K.I.); 3Department of Emergency Medicine, Juntendo University Nerima Hospital, Tokyo 177-8521, Japan; y-hotchi@juntendo.ac.jp; 4Department of Gastroenterology, Shin-Yurigaoka General Hospital, Kawasaki 215-0026, Japan

**Keywords:** portal hypertension, portal hemodynamics, liver impairment, splenic embolization, hypertensive gastropathy

## Abstract

This study investigated the impact of partial splenic embolization (PSE) on portal hypertensive gastropathy (PHG). We retrospectively analyzed endoscopic findings and the portal venous system of 31 cirrhotic patients with PHG. The improved group was defined as the amelioration of PHG findings using the McCormack classification. Child–Pugh scores of the improved group (18 of 31 patients) were significantly lower compared with those of the non-improved group (*p* = 0.018). The changes in the diameters of the portal trunk and those of the spleno-portal junction and spleen hilum in the splenic vein of the improved group were significantly larger than those of the non-improved group (*p* = 0.007, *p* = 0.025, and *p* = 0.003, respectively). The changes in the diameters of the portal vein and splenic hilum of the splenic vein showed significant correlations with Child–Pugh score (r = 0.386, *p* = 0.039; r = 0.510, *p* = 0.004). In a multivariate analysis of baseline factors related to the improved group, Child–Pugh grade A was significantly associated with the improvement of PHG (odds ratio 6.875, *p* = 0.033). PSE could be useful for PHG, especially in patients with Child–Pugh grade A, at least in the short term.

## 1. Introduction

Portal hypertensive gastropathy (PHG) is a gastric mucosal lesion in patients with portal hypertension (PH). It is characterized by vasodilation of the mucosal capillaries and gastric submucosal veins without inflammatory change [1]. Hyperdynamic circulation caused by portal hypertension results in gastric mucosal blood flow congestion and edema [2]. PHG is a common cause of anemia and upper gastrointestinal bleeding in cirrhotic patients [3,4]. Although the prevalence of PHG in patients with cirrhosis ranges from 20% to 80%, only 10–15% of patients with PHG develop overt gastric bleeding [5,6]. 

Pharmacological therapies such as β-blockers, vasopressin, somatostatin, transjugular portosystemic shunt (TIPS), partial spleen embolization (PSE), and argon plasma coagulation (APC) at the bleeding state are promising tools to control PHG [4,7,8,9]. PSE is a minimally invasive procedure that was developed to treat hypersplenism due to chronic hepatic disease. It causes ischemic necrosis of the splenic tissue through the embolization of the splenic artery branch and the reduction in portal vein pressure through downregulation of the splenic venous return. Thus, PSE has been proven to be a safe and effective treatment procedure for portal hypertension in liver cirrhosis [10]. In the clinical setting, PSE has been applied to elevated platelet count and has been simultaneously or heterochronously combined with balloon-occluded retrograde transvenous obliteration (B-RTO), which has resulted in the improvement of liver function [10,11].

There are few reports regarding endoscopic findings before and after PSE in cirrhotic patients and the application of PSE for the treatment of PHG. Therefore, the purpose of this study was to investigate the therapeutic effect of PSE for PHG in relation to portal hemodynamics.

## 2. Materials and Methods

### 2.1. Patients and Treatments

This was a single-facility, retrospective study conducted at Juntendo University Nerima Hospital. The study was approved by the ethical review board of Juntendo University Nerima Hospital (No. E21-0259) and was conducted in accordance with the 1964 Declaration of Helsinki and its later amendments. Written informed consent was obtained from each patient before the procedure. The flowchart of this study is shown in Figure 1. A total of 65 patients with liver cirrhosis were treated with PSE between April 2006 and November 2021 at our institution. Among these patients, 32 were excluded because of a lack of upper gastrointestinal endoscopy both before and after PSE. Among the remaining 33 patients, 2 were excluded because of the lack of PHG findings. Finally, this study involved 31 patients.

### 2.2. Endoscopy and Classification of Portal Hypertensive Gastropathy

Upper gastrointestinal endoscopic examination was performed within 4 weeks before PSE and 1–8 weeks after PSE. PHG was diagnosed by upper gastrointestinal endoscopic examination in all patients. The classification of PHG was evaluated according to the McCormack criteria (1). Specifically, PHG findings were classified as “Mild”, with features such as fine pink speckling (M1), superficial reddening (M2), and a snakeskin (mosaic) appearance (M3), and “Severe” with features such as cherry red spots (S1) or diffuse hemorrhagic lesions (S2). In this study, the PSE-improved group was defined as “Severe” (S1 or S2) to “Mild”, or M3, or M2 to M1, or absent. The McCormack classifications were evaluated independently by three researchers (H.O., Y.F., and E.N.) who have more than 15 years of experience in upper gastrointestinal endoscopy. Because 27 of 62 cases (44%) of endoscopic findings were not found in a unanimous decision, a majority decision was finally adopted.

### 2.3. Partial Splenic Embolization

Splenic arteriography was performed using a biplane digital subtraction angiography system (Siemens Artis Q, Siemens Healthcare, Erlangen, Germany) via right femoral artery access. Using a 4-Fr catheter (Cobra; Terumo Clinical Supply, Gifu, Japan), a contrast agent (Iomeron 300; Eisai, Tokyo, Japan) was administered at a dose of 15 cc and an injection rate of 3 cc/s. A 2.0-Fr microcatheter (Bobsled, Piolax, Yokohama, Japan) was used to approach the spleen vessels. To embolize 60–80% of the spleen volume, peripheral branches were selectively cannulated and embolized by approximately 2 mm gelatin sponge cubes. Peri-interventional antibiotic prophylaxis was administered with broad-spectrum antibiotics before PSE and for 1 week post-PSE. Contrast-enhanced computed tomography (CT) was performed to identify the embolic volume of the spleen and the incidence of complications.

### 2.4. Computed Tomography Imaging Analysis

Contrast-enhanced CT was undertaken with a CT system (Aquilion 1; Canon Medical Systems, Tochigi, Japan) 1–4 weeks before PSE and within 1 week after PSE. After pre-contrast scanning, iodinated contrast materials (Iomeron 300; Eisai, Tokyo, Japan) were injected at an injection duration of 30 s. The arterial dominant phase was determined by the bolus-tracking method as the time at which the abdominal aorta CT value reached a level of 200 HU plus 17 s. Subsequently, a 70 s delay for the port venous phase and a 120 s delay for the equilibrium phase were imaged. Coronal CT images of the port venous phase were displayed on the zaiostation 2™ (Ziosoft, Tokyo, Japan), and diameters of the portal trunk caudal cross-section of the Spiegel lobe, the root of the left gastric vein, and the splenic vein at the spleno-portal junction and splenic hilum corresponding the most caudal part of pancreas tail were measured. Splenic volumes, including embolic volumes, were also measured. Blind analyses of the CT images were undertaken by two radiologists (D.T. and Y.O.) who have more than 10 years of experience in the interpretation of abdominal CT imaging. The average value measured by the two radiologists was adopted for analysis of the CT images.

### 2.5. Statistical Analysis

Continuous variables are expressed as median values (range) and were analyzed using the Mann–Whitney U test. Wilcoxon signed-rank tests were used in cases of paired data. Fisher’s exact test was used to compare categorical data. Spearman’s rank correlation coefficients were performed to determine the associations between pairs of variables. Variables of pretreatment factors with *p*-values of <0.10 in the univariate analysis were re-evaluated by multiple logistic regression analysis using the backward selection method of likelihood ratio to identify the factors associated with the improvement of PHG by PSE. All tests were two-sided, and *p*-values of <0.05 were considered to be statistically significant. All statistical analyses were performed using SPSS Statistics for Windows, version 27 (IBM Corp., Armonk, NY, USA).

## 3. Results

### 3.1. Baseline Characteristics

Table 1 summarizes the baseline characteristics of all patients included in this study. This study compromised 31 patients (15 males and 16 females) aged 29 to 78 years. The main purpose of PSE was as follows: treat PHG in ten patients, increase inplatelet count in six patients, heterochronously combine PSE and B-RTO or endoscopic injection sclerotherapy in thirteen patients, and improve hepatic impairment in two patients. Among the thirty-one patients, nine patients (29%) had atrophic gastritis in the endoscopic findings. The Child–Pugh score was five in five patients, six in eight patients, seven in ten patients, eight in five patients, and ten in three patients. There were no patients with Child–Pugh class C. Ten patients had ascites and twenty-one had none. Variceal formation such as esophageal and/or gastric varices was found in twenty-eight patients, but not in three patients. Only one patient took non-selective beta-blockers. Four patients required blood transfusions and six patients took oral iron supplementation prior to PSE. There were no significant correlations between the McCormack classification and albumin–bilirubin (ALBI) grade or splenic volume (r = 0.203, *p* = 0.273; and r = 0.028, *p* = 0.881, respectively).

### 3.2. The Effect of Partial Splenic Embolization for Portal Hypertensive Gastropathy

To determine whether PSE improved PHG in patients with portal hypertension, we assessed the endoscopic findings of PHG before and after PSE. Regarding the blind reading of endoscopy findings, a unanimous decision was reached for 4 of 5 cases with severe, but not for 26 of 59 cases with mild PHG. As illustrated in Figure 2, the McCormack classification was significantly improved after PSE (*p* < 0.001). PHG was improved in 18 of 31 (58%) patients (improved group). In contrast, PHG was not improved in 13 of 31 (42%) patients (non-improved group). In particular, all five severe McCormack cases improved to mild PHG. Specifically, among three McCormack S1 cases, one patient had improved M1, and two had M2, while among McCormack S2, two cases had improved M1 and M3. Moreover, among eight of seventeen cases with McCormack S3, improved M1 was observed in six patients and absent in two patients. After the PSE procedure, there was no need for blood transfusion and/or iron supplementation in six patients who required blood transfusions and/or iron supplementation prior to PSE. Regarding complications, one patient developed a splenic vein thrombus after PSE. The other patients had no complications related to PSE. Comparisons of each variable between the PHG-improved group and the non-improved group are shown in Table 2. Child–Pugh scores of the improved group were significantly lower compared with those of the non-improved group (*p* = 0.018). There were no significant differences in the other pretreatment parameters between the two groups.

### 3.3. Multivariate Analysis Factors Associated with the Amelioration of Portal Hypertensive Gastropathy

As shown in Table 3, univariate and multivariate logistic regression analyses of baseline factors related to the amelioration of PHG were performed. Univariate logistic regression analysis revealed that Child–Pugh grade A and a platelet count of 6 × 10^8^/L or more were related to the amelioration of PHG (*p* = 0.003 and *p* = 0.048, respectively). In the multiple logistic regression analysis, Child–Pugh grade A classification was a significant predictor associated with the amelioration of PHG (odd ratios 6.875, *p* = 0.033). As listed in Appendix A, with regards to the amelioration rate of PSE for PHG, the rates of patients with Child–Pugh scores of 5, 6, 7, 8, and 9 were 83%, 83%, 50%, 50%, and 0%, respectively.

### 3.4. Radiological Changes before and after Partial Spleen Embolization

Radiological changes before and after partial spleen embolization are shown in Figure 3. There was no significant difference in the embolic volume of the spleen by PSE between the improved group and the non-improved group (*p* = 0.679). Changes in the portal trunk and both the spleno-portal junction and splenic hilum of the splenic vein of the improved group were significantly larger than those of the non-improved group (*p* = 0.007, *p* = 0.025, and *p* = 0.003, respectively), whereas those in the left gastric vein of the improved group tended to be larger compared with those of the non-improved group (*p* = 0.062). There was no significant relationship between the embolic volume of the spleen and changes in the hilum of the splenic vein, which is adjacent to the spleen (r = −0.155, *p* = 0.413).

### 3.5. The Relationship between Changes in Portal Hemodynamics after PSE and Child–Pugh Score

As illustrated in Figure 4, changes in the diameters of the portal vein and the splenic hilum of the splenic vein before and after PSE showed significant correlations with Child–Pugh score (r = 0.386, *p* = 0.039; r = 0.510, *p* = 0.004), whereas there was no significant correlation between changes in the left gastric vein and the splenic vein at the spleno-portal junction and Child–Pugh score (r = 0.270, *p* = 0.142; r = 0.294, *p* = 0.115, respectively).

## 4. Discussion

The present study showed that Child–Pugh A was an independent and significant factor for ameliorating PHG by PSE. Additionally, we found that radiological changes in the portal vein and splenic hilum of the splenic vein after PSE were correlated with Child–Pugh score based on hemodynamic analysis.

PHG is a common endoscopic finding in chronic liver disease. In previous studies, the severity of PHG was related to the duration and severity of chronic liver disease [12,13]. Nevertheless, liver function was not associated with PHG severity in the present study. One possible explanation for this difference is the limited criteria of PSE that excluded patients with Child–Pugh C in our study. Although the therapeutic indication of PSE for PHG is still controversial, patients with upper gastrointestinal bleeding and/or progression of anemia would obviously require treatment intervention. There are several treatment options for PHG such as pharmacological therapy, APC, PSE, splenectomy, and TIPS [4,7,8,9,13,14]. Nevertheless, it is not obvious which therapy is effective for PHG. Additionally, because TIPS is not supported by health insurance in Japan [15], therapeutic options for PHG are limited.

The spleen is a regulatory organ that maintains portal vein flow into the liver. PSE was developed for primary and secondary hypersplenism following the report by Spigos et al. using antibiotic coverage and pain control after PSE [16]. Several reports, including a case report, showed that PSE has beneficial effects on PHG [4,14,17]. Previously, Ohmagari et al. reported that 71% of patients with PHG had improved following PSE in a comparative study of 17 cases with PSE and 13 cases without PSE [14]. A recent study demonstrated the efficacy of PSE in seven patients with recurrent PH-induced upper gastrointestinal bleeding from PHG refractory to TIPS [4]. In the current study, we clearly demonstrated the efficacy of PSE for 31 patients with PHG, with a high control rate of 100% in severe PHG cases. However, Anegawa et al. investigated the effect of laparoscopic splenectomy on PHG in 49 cirrhotic patients and found that PHG was improved in 94% of patients with severe PHG and 38% with mild PHG [13]. Splenectomy decreases portal pressure and inflow to the portal venous system. Similarly, PSE can have an effect on the reduction in portal pressure to greater or lesser degrees. Our result of the amelioration rate by PSE of 100% in severe PHG and 50% in mild PHG is relatively in line with the report by Anegawa et al. The development of portal vein thrombosis is inevitable after splenectomy [18,19]. In contrast, PSE had an extremely low rate of complications in the present study, with only one case of splenic vein thrombosis. Although severe PHG or M3 PHG in the McCormack criteria were not significantly predictable factors related to the improvement of PHG, it is noteworthy that all severe PHG cases improved in the present study. Furthermore, considering that there was no need for blood transfusions and/or iron supplementation in all patients after PSE, the procedure could contribute to bleeding control due to PHG.

The combination of pharmocological and endoscopic therapy is considered first-line therapy in acute bleeding situations resulting from portal hypertension [8]. TIPS is the treatment of choice for first-line treatment failure. However, in the clinical setting, there is an urgent need to manage PHG bleeding. PSE has various characteristics such as elevated platelet count, less invasiveness, rapid response to portal pressure, improvement of liver function, and the preservation of some splenic function [20,21,22]. Furthermore, based on our results, PSE could be a therapeutic option in cases of gastrointestinal bleeding by PHG without variceal bleeding.

Hepatic venous pressure gradient (HVPG) measurement is the gold-standard method to assess the presence of PH [8]. Ishikawa et al. reported that the splenic non-infarction volume was associated with a clinically significant HVPG response to PSE in cirrhotic patients with hypersplenism [23]. As a matter of course to improve PHG, adequate reduction in portal hypertension by PSE is required. Nevertheless, HVPG was not investigated in our study. If HVPG measurement was performed during PSE, the optimal embolic rate of the spleen would be easy to determine. Contrary to Ishikawa’s report, embolic volume was not associated with the efficacy of PSE for PHG in the present study because of our small cohort, which excluded Child–Pugh C.

PH develops because of an elevation in intrahepatic vascular resistance caused by the progression of chronic liver disease [24]. In a normal liver, venous return from visceral organs joins the portal vein trunk and flows into the liver. The progression of PH exposes some visceral organs to this congestion. In terms of the portal hemodynamics of PHG, gastric venous flow, such as short gastric vein and posterior gastric vein, are likely to be susceptible to the effects of PSE. Thus, we must include suitable patients with PHG who would benefit from PSE. To achieve this goal, we analyzed portal hemodynamics according to liver function before and after PSE. Our results suggest that PSE has an advantage in the improvement of PHG if it was performed in patients with good liver function, such as those with Child–Pugh grade A. In other words, the portal vein system is likely to be affected by PSE in patients with good liver function. If the flexibility of the portal vein system might be maintained, the extent of the reduction in splenic vein flow by PSE may affect the pressure of the portal vein system. Although, patients with severe liver function, such as those with Child–Pugh class C, were excluded from this study because of a high incidence of complications [25]. PSE may not be necessarily recommended for PHG in patients with poor liver function, such as those with a Child–Pugh score of 9 or more.

Our analysis also suggests that whether PSE efficiently affects the portal system, including the portal trunk, splenic vein, and left gastric vein, is essential for the success of PSE for PHG. Interestingly, the left gastric vein, which is often derived from the splenic vein, was not significantly influenced by PSE, and the changes in vessel diameter were not correlated with Child–Pugh score. These results indicate the possibility of a different etiology in the incidence of PHG without portal venous flow. Several recent studies demonstrated that the occurrence of PHG was related not only to increased PH but also to the activation of cytokines, growth factors, and hormones that are involved in hyperdynamic gastric circulation [26,27,28].

Portal hemodynamics in patients with PH is complicated. Patients with good liver function might have the plasticity of the portal vein, which leads to reduced portal hemodynamics by PSE. In contrast, patients with severe liver impairment, who often have a hepato-fugal flow of the portal vein [29], might be less susceptible to splenic venous return by PSE. As a matter of fact, the efficacy of PSE for PHG was 0% in patients with a Child–Pugh score of 9 in our cohort, although PSE was not performed in patients with Child–Pugh grade C. To our knowledge, this is the first report to indicate endoscopic changes of PHG by PSE based on portal hemodynamics.

This study has several limitations. The first is the difficulty of an endoscopic diagnosis of PHG. Because it is difficult to differentiate redness on the body of the stomach resulting from *Helicobacter pylori* infection from that of PHG [30], the endoscopic diagnosis of PHG, especially in mild-stage disease, might result in intra-observer differences. Nevertheless, it is noteworthy that unanimous decisions were reached for 4 of 5 severe McCormack cases. Second, this was a short-term small cohort study. Long-term and larger studies are needed to confirm the efficacy of PSE for PHG. Third, we did not investigate HVPG during the PSE procedure. Fourth, we only measured the vessel diameters of the portal system and did not estimate the direction of the portal flow. Nevertheless, the strengths of the present study include the detailed assessment of the portal venous system before and after PSE for PHG.

## 5. Conclusions

This study showed that PSE could initiate a change in PHG findings. PSE may be a useful procedure for PHG, especially in patients with Child–Pugh grade A, at least in the short term.

## Figures and Tables

**Figure 1 jcm-12-02662-f001:**
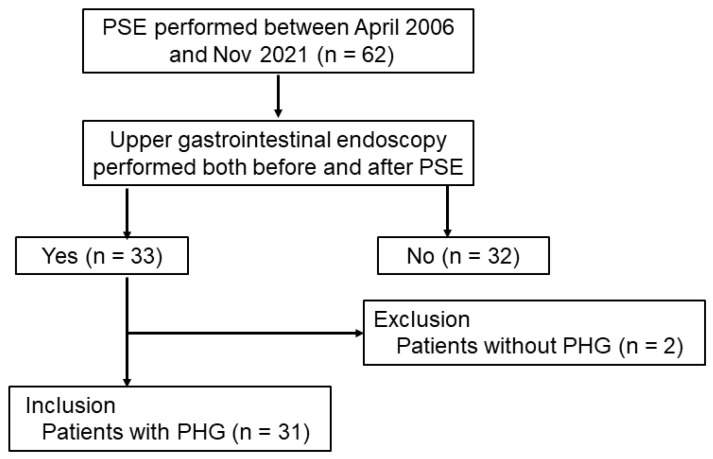
Study design and inclusion and exclusion criteria for partial spleen embolization.

**Figure 2 jcm-12-02662-f002:**
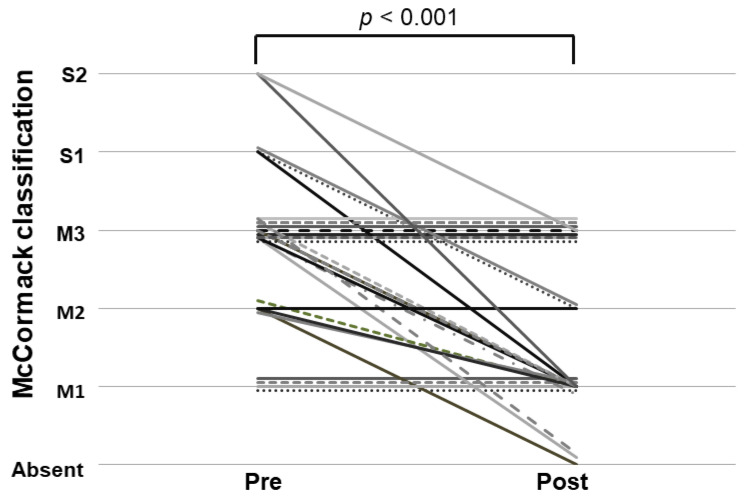
McCormack classification of endoscopy before and after partial spleen embolization.

**Figure 3 jcm-12-02662-f003:**
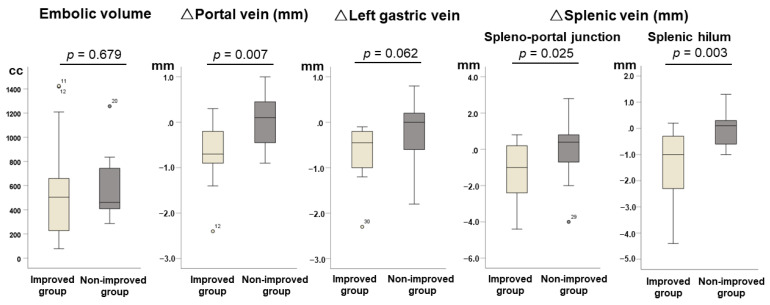
Radiological changes before and after partial spleen embolization.

**Figure 4 jcm-12-02662-f004:**
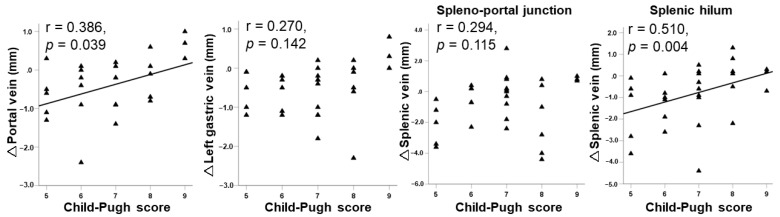
The relationship between changes in portal hemodynamics after partial splenic embolization and Child–Pugh score.

**Table 1 jcm-12-02662-t001:** Patients’ characteristics.

	All Patients
	N = 31
Age, years (range)	61 (29–78)
Sex, male/female	15/16
Etiology, HBV/HCV/ALC/NASH/others	2/12/8/2/7
Purpose, PHG/platelet increase/additional treatment of EIS or BRTO/improvement of hepatic impairment	10/6/13/2
White blood cells, 10^6^/L	2600 (1100–6300)
Hemoglobin, g/dL	11.3 (7.5–14.4)
Platelets, ×10^4^/μL	4.7 (2.1–22.4)
Prothrombin activity, %	63 (34–99)
Total bilirubin, mg/dL	1.1 (0.3–2.3)
Aspartate aminotransferase; median, IU/L (range)	45 (15–334)
Alanine aminotransferase; median, IU/L (range)	31 (9–317)
Albumin, g/dL	3.4 (2.1–4.8)
Ascites, with/without	10/21
Variceal formation, with/without	28/3
Ammonia, μg/dL	78 (19–210)
Child–Pugh score, 5/6/7/8/9	5/8/10/5/3
ALBI score (range)	−2.08 (−3.27–−0.91)
mALBI grade	8/5/14/4
McCormack criteria, M1/M2/M3/S1/S2	4/5/17/3/2
Embolic volume, cc (range)	492.5 (78.7–1425)
Atrophic gastritis, yes/no	10/21

Median (range) or n; ALBI, albumin bilirubin; BRTO, balloon occluded retrograde transvenous obliteration; EIS, endoscopic injection sclerotherapy; mALBI, modified albumin–bilirubin; McCormack M1, fine pink speckling; M2, superficial reddening; M3, snakeskin (mosaic) appearance; S1, cherry red spots; S2, diffuse hemorrhagic lesion.

**Table 2 jcm-12-02662-t002:** A comparison of each variable between the portal hypertensive gastropathy-improved group and the non-improved group.

	Improved Group	Non-Improved Group	
	n = 18	n = 13	*p*-Value
Age, years	64.5 (56.5–73)	60 (49–71.5)	0.258
Sex, male/female	7/11	8/5	0.656
Etiology, HBV/HCV/ALC/NASH/others	1/5/6/2/4	1/7/2/0/3	
White blood cells, 10^6^/L	2650 (1850–3525)	2600 (2150–3550)	0.921
Hemoglobin, g/dL	11.25 (9.175–12.325)	11.5 (9.7–12.4)	0.718
Platelets, ×10^4^/μL	5.15 (3.225–8.025)	4.6 (3.45–5.2)	0.441
Prothrombin activity, %	65 (55–73.5)	57 (54.5–67)	0.352
Total bilirubin, mg/dL	1.05 (0.7–1.4)	1.5 (0.75–1.7)	0.261
AST, IU/L	39 (27.5–54.25)	46 (38.5–53.5)	0.089
ALT, IU/L	28 (21.5–58.75)	34 (26–63.5)	0.465
Albumin, g/dL	3.6 (3.375–3.925)	3.3 (2.85–3.85)	0.830
Ascites, with/without	4/14	6/7	0.247
Variceal formation, with/without	16/2	12/1	1.000
Ammonia, μg/dL	73.5 (55–103.5)	91 (58.5–123)	0.226
Child–Pugh score	6 (5–7)	7 (7–8.5)	0.018
ALBI score	−2.51 (−3.11–−1.64)	−1.94(−2.535–−1.465)	0.352
mALBI grade 1/2a/2b/3	6/3/8/1	2/2/6/3	0.743
McCormack criteria M1/M2/M3/S1/S2	0/4/9/3/2	4/1/8/0/0	0.313
Portal vein, mm	13 (11–14.7)	12.5 (10.08–14.5)	0.777
Left gastric vein, mm	6.1 (4.0– 9.3)	4.4 (3.73 –5.25)	0.161
Splenic vein at spleno-portal junction, mm	10.7 (8.5–13.4)	9.95 (6.98–12.48)	0.267
Splenic vein at splenic hilum, mm	11.6 (8.3–16.0)	9.4 (7.0–13.0)	0.183

Median (Q1–Q3) or n; AST, aspartate aminotransferase; ALC, alcohol; ALT, alanine aminotransferase; HBV, hepatitis B virus; HCV, hepatitis C virus; NASH, non-alcoholic steatohepatitis; McCormack M1, fine pink speckling; M2, superficial reddening; M3, snakeskin (mosaic) appearance; S1, cherry red spots; S2, diffuse hemorrhagic lesion.

**Table 3 jcm-12-02662-t003:** Univariate and multivariate logistic regression analyses of baseline factors related to the improvement of portal hypertensive gastropathy by partial splenic embolization.

	Univariate Analysis			Multivariate Analysis		
	OR	95% CI	*p*-Value	OR	95% CI	*p*-Value
Age, ≥60 years	1.250	0.283–5.525	0.769			
Sex, male	0.686	0.164–2.874	0.606			
White blood cells, ≥3000/μL	0.800	0.181–3.536	0.769			
Hemoglobin, ≥11 g/dL	0.781	0.183–3.342	0.739			
Platelets, ≥6.0 × 10^4^/μL	9.600	1.020–90.343	0.048	6.437	0.623–66.544	0.118
Total bilirubin, ≥1.3 mg/dL	0.429	0.099–1.857	0.257			
AST, ≥45 IU/L	0.990	0.9471–1.010	0.334			
ALT, ≥31 IU/L	0.991	0.973–1.009	0.328			
Albumin, ≥3.5 g/dL	3.536	0.780–16.032	0.102			
Prothrombin activity, ≥70%	0.364	0.060–2.194	0.270			
Ascites, with	0.333	0.070–1.581	0.167			
Variceal formation, with	0.667	0.054–8.240	0.752			
Ammonia, ≥70 μg/dL	0.556	0.124–2.419	0.443			
Child–Pugh grade A, yes	6.875	1.171–40.378	0.033	6.875	1.171–40.378	0.033
McCormack, M3 or severe	2.187	0.452–10.576	0.330			

ALT, alanine aminotransferase; AST, aspartate aminotransferase; CI, confidence interval; OR, odds ratio.

## Data Availability

Data are contained within the article and the Appendix A.

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
