# Peer review of "The Impact of Partial Splenic Embolization on Portal Hypertensive Gastropathy in Cirrhotic Patients with Portal Hypertension"

_jcm, 2023, doi:10.3390/jcm12072662_

Round 1
Reviewer 1 Report
Thank you for submitting your manuscript to our journal. I have several questions for you, although this study has demonstrated new approach to evaluate therapeutic effect by PSE (partial splenic embolization) on PHG (poral hypertensive gastropathy).
1. What do you think PSE has the most clinical impact on? You mentioned Child-Pugh A patients would benefit from PSE regarding endoscopic findings of gastropathy. What are your other thoughts for advantage of having PSE? Is there increased number of platelets, or prolonged time to decompression? Please describe the advantage of PSE in clinical aspects.
2 The use of beta blockers is considered an important component of the management of management of portal hypertension, recommended by clinical guidelines. Please describe the use of beta blockers. Also, please advise that there is need or to mention that factors related to portal hypertension include ascites and variceal formation.
3 Portal pressure gradient management is typically performed by TIPS, if other pharmacological intervention such as beta-blockers fails to respond. What do you think of PSE lowering PPG, instead of the vessels flow?
Author Response
#1. Reviewer1
Thank you for submitting your manuscript to our journal. I have several questions for you, although this study has demonstrated new approach to evaluate therapeutic effect by PSE (partial splenic embolization) on PHG (poral hypertensive gastropathy).
- What do you think PSE has the most clinical impact on? You mentioned Child-Pugh A patients would benefit from PSE regarding endoscopic findings of gastropathy. What are your other thoughts for advantage of having PSE? Is there increased number of platelets, or prolonged time to decompression? Please describe the advantage of PSE in clinical aspects.
Response: We thank the reviewer for this clinically important comment. In the present cohort, there was no need for blood transfusions and/or iron supplementation after PSE in six patients who required blood transfusions and/or iron supplementation prior to PSE. This information has been added to the Results section of the revised manuscript (lines 137–138, and 160–161). We also emphasized that PSE could contribute to bleeding control due to PHG in the revised manuscript (lines 254–256). Moreover, several reports indicated that PSE has beneficial effects not only on elevation of platelet count but also on liver functional improvement in cirrhotic patients. We have added this to the Discussion section of the revised manuscript (lines 260–262).
- The use of beta blockers is considered an important component of the management of management of portal hypertension, recommended by clinical guidelines. Please describe the use of beta blockers. Also, please advise that there is need or to mention that factors related to portal hypertension include ascites and variceal formation.
Response: We thank the reviewer for highlighting this issue. Because only one patient took beta blockers in the present cohort, we have included this information in the revised manuscript (line 135). However, additional data was generated to address the situation with or without ascites and variceal formation in the revised manuscript (lines 133–135).
3 Portal pressure gradient management is typically performed by TIPS, if other pharmacological intervention such as beta-blockers fails to respond. What do you think of PSE lowering PPG, instead of the vessels flow?
Response: Thank you for this clinically important comment. The combination of pharmocological and endoscopic therapy is considered as first-line therapy in acute bleeding situations because of portal hypertension. Transjugular intrahepatic portosystemic shunt (TIPS) is the treatment of choice for first-line treatment failure. However, in the clinical setting, there is an urgent need to manage PHG bleeding. PSE has various characteristics such as elevation of platelet count, less invasiveness, rapid response to portal pressure, improvement in liver function, and preservation of some splenic function. Furthermore, based on our results, PSE could be a therapeutic option in cases of gastrointestinal bleeding by PHG without variceal bleeding, although PSE does not have a direct effect on reduction of portal pressure. These discussions have been added to the revised manuscript (lines 258–264).
Reviewer 2 Report
Dear Editor,
In this retrospective study, the Authors aimed to investigate the potential benefit of partial splenic embolization in a cohort of cirrhotic patients (Child-Pugh A or B) with portal hypertension suffering from hypertensive gastropathy. Saeki et al. identified 31 patients undergoing PSE and they compared radiological and endoscopic changes pre- and post-PSE, demonstrating a good endoscopic response in patients with preserved liver function with severe gastropathy with a relatively low risk of complications (one case of splenic vein thrombosis).
However, the work has some serious shortcomings.
Firstly, the strongest outcome of the treatment is not even mentioned: that is, the reduction in PHG-bleeding. The authors should mention whether there was a reduction in the need for blood transfusions or (as surrogate marker) for iron supplementation even in view of the initial Hb level of the treated patients (median Hb value of 11.3 g/dL). Moreover, endoscopic improvement, considering the high heterogeneity of observation among endoscopists, is a weak point and should be stressed deeply (mainly because only 5 patients were classified with severe gastropathy).
Secondly, when selecting patients, it must be specified whether they could not be candidates for TIPS placement as TIPS is an effective treatment (see cit. 21) and how many of them were potentially eligible for liver transplantation. Furthermore, the absence of the HVPG measurement makes it necessary that additional data such as the presence of oesophageal varices and/or concomitant treatment with beta-blockers should be reported.
Other considerations:
- Follow-up (with further endoscopic investigations) should be prolonged to examine whether the initial improvement has been maintained over time.
- The discussion section is chaotic and should be made more clear.
- Line 39: better explain the PSE principle and its application areas with relevant citations.
- Line 42: citation number 3 does not correspond to what is stated in the text. Please also specify the result of the reports mentioned.
- Line 57: age and gender should be reported in the results, as they are not methods of patient selection in this case.
- Lines 119-122: the meaning is not clear and should be reformulated.
- Line 139: please correct or specify the term PHS.
- Line 141: please specify the extent of the endoscopic improvement in accordance with Figure 2 (e.g. how many patients from S2 to M3, from M3 to M1, etc.)
- Table 2: it would appear from what is reported that there are no patients in the 'improved group” with PHG graded S1 or S2, in disagreement with what is reported in the text.
- Please report the value of platelets in the same unit in different tables.
- The term “hypertensive gastropathy” could be added as a keyword.
Author Response
#2. Reviewer 2
In this retrospective study, the Authors aimed to investigate the potential benefit of partial splenic embolization in a cohort of cirrhotic patients (Child-Pugh A or B) with portal hypertension suffering from hypertensive gastropathy. Saeki et al. identified 31 patients undergoing PSE and they compared radiological and endoscopic changes pre- and post-PSE, demonstrating a good endoscopic response in patients with preserved liver function with severe gastropathy with a relatively low risk of complications (one case of splenic vein thrombosis).
However, the work has some serious shortcomings.
Firstly, the strongest outcome of the treatment is not even mentioned: that is, the reduction in PHG-bleeding. The authors should mention whether there was a reduction in the need for blood transfusions or (as surrogate marker) for iron supplementation even in view of the initial Hb level of the treated patients (median Hb value of 11.3 g/dL). Moreover, endoscopic improvement, considering the high heterogeneity of observation among endoscopists, is a weak point and should be stressed deeply (mainly because only 5 patients were classified with severe gastropathy).
Response: Thank you for your insightful comments. In the present cohort, there was no need for blood transfusions and/or iron supplementation after PSE in six patients who required blood transfusions and/or iron supplementation prior to PSE. This information has been reported in the Results section of the revised manuscript (lines 135–136). Based on this result, we also emphasized the clinical impact of PSE on PHG in cirrhotic patients in the Discussion section of the revised manuscript (lines 254–256).
Both McCormack classification and Toyonaga classification, the other classification system for PHG, were included in the original manuscript. Therefore, in the revised manuscript, we recalculated the results of blind reading by McCormack classification alone. Consequently, a unanimous decision was reached for 4 of 5 cases with severe PHG, but not for 26 of 59 cases with mild PHG. Taken together, we believe that endoscopic assessment of PHG in the present cohort was relatively reliable, as described in the limitations subsection. These contents had been included in the revised manuscript (lines 150–152).
Secondly, when selecting patients, it must be specified whether they could not be candidates for TIPS placement as TIPS is an effective treatment (see cit. 21) and how many of them were potentially eligible for liver transplantation. Furthermore, the absence of the HVPG measurement makes it necessary that additional data such as the presence of oesophageal varices and/or concomitant treatment with beta-blockers should be reported.
Response: Thank you for bringing this problem to our attention. Because TIPS is not supported by health insurance in Japan, it is not a popular procedure for portal hypertension, unlike in other countries. Accordingly, PSE is a useful option for PHG. Regarding potentially eligible candidates for liver transplantation, the MELD scores of 28 patients were 9 or less and those of 3 patients were 11 to 17. Patients with Child–Pugh C were also excluded from our study. Hence, we believe that patients who were overtly eligible for liver transplantation were not included in the present cohort. We have added a sentence about the current status of TIPS in Japan to the revised manuscript (lines 229–230).
In fact, the other reviewer also recommended presenting the presence of esophageal varices and/or concomitant treatment with beta blockers. In accordance with these comments, we have added additional data including the presence of variceal formation. Because only one patient took beta blockers in the present cohort, we could not analyze the effect of beta blockers. We have included this information in the revised manuscript (lines 133–135 and Tables 1–3).
Hepatic venous pressure gradient (HVPG) measurement is the gold-standard method to assess the presence of PH. Because PSE involves trans-arterial access, we did not investigate HVPG via transjugular access. If HVPG measurement was performed during PSE for PHG, the optimal embolic rate would be easy to determine. We have added these comments to the revised manuscript (lines 272–273).
Other considerations:
- Follow-up (with further endoscopic investigations) should be prolonged to examine whether the initial improvement has been maintained over time.
Response: Unfortunately, we could obtain no satisfactory data of prolonged endoscopic findings. We are aware that this is a weak point of our study, as described in the limitations subsection. Therefore, we have added the phrase “at least in the short term” to the Conclusions and abstract of the revised manuscript (line 24, and lines 324–325).
- The discussion section is chaotic and should be made more clear.
Response: In accordance with the reviewer’s comment, the Discussion section has been rewritten.
- Line 39: better explain the PSE principle and its application areas with relevant citations.
Response: Thank you for your helpful comment. We have added the principle of PSE and its application to the revised manuscript (lines 40–46).
- Line 42: citation number 3 does not correspond to what is stated in the text. Please also specify the result of the reports mentioned.
Response: Thank you for pointing out this error. To our knowledge, there are no reports regarding endoscopic findings before and after PSE in cirrhotic patients and the application of PSE for managing PHG. Thus, we have deleted this notation.
- Line 57: age and gender should be reported in the results, as they are not methods of patient selection in this case.
Response: Thank you for your useful comment. Age and sex have been reported in the Results section of the revised manuscript (line 125).
- Lines 119-122: the meaning is not clear and should be reformulated.
Response: In accordance with the reviewer’s comment, we have changed the notation as follows: “heterochronous combination of PSE and B-RTO or endoscopic injection sclerotherapy.”
- Line 139: please correct or specify the term PHS.
Response: Thank you pointing out this error. We have changed the notation, which is now as follows: PHG.
- Line 141: please specify the extent of the endoscopic improvement in accordance with Figure 2 (e.g. how many patients from S2 to M3, from M3 to M1, etc.)
Response: Thank you for your useful comment. For the reader’s convenience, we have added the specific data of McCormack classification after PSE to the revised manuscript (lines 156–161).
- Table 2: it would appear from what is reported that there are no patients in the 'improved group” with PHG graded S1 or S2, in disagreement with what is reported in the text.
Response: This is a mistake in the McCormack criteria of Table 2. We have swapped the improved group and non-improved group.
- Please report the value of platelets in the same unit in different tables.
Response: In accordance with the reviewer’s comment, we have used the same unit for platelets in all tables.
- The term “hypertensive gastropathy” could be added as a keyword.
Response: Thank you for your helpful comment. We have added “hypertensive gastropathy” as a keyword to the revised manuscript (lines 25–26).
Round 2
Reviewer 2 Report
Dear Editor,
the authors have adequately answered the questions raised. However, some minor language mistakes persist and need correction.
Author Response
The reviewer referred quality of English language in the revised manuscript.
Both original and revised manuscripts had been checked by a native speaker of English from a company providing professional English editing services as described in the acknowledgments.
Although the reviewer dose not refer to specific errors, several minor language mistakes have been corrected in the revised manuscript.